# A Degradable and Osteogenic Mg-Based MAO-MT-PLGA Drug/Ion Delivery System for Treating an Osteoporotic Fracture

**DOI:** 10.3390/pharmaceutics14071481

**Published:** 2022-07-16

**Authors:** Changxin Liu, Wen Zhang, Ming Gao, Ke Yang, Lili Tan, Wei Zhao

**Affiliations:** 1School of Materials Science and Engineering, University of Science and Technology of China, No. 72 Wenhua Road, Shenyang 110016, China; cxliu19b@imr.ac.cn (C.L.); wzhang19b@imr.ac.cn (W.Z.); 2Shi-Changxu Innovation Center for Advanced Materials, Institute of Metal Research, Chinese Academy of Sciences, No. 72 Wenhua Road, Shenyang 110016, China; mgao16b@imr.ac.cn (M.G.); kyang@imr.ac.cn (K.Y.); 3Department of Orthopedics, The Fourth Hospital of China Medical University, No. 77 Puhe Road, Shenyang 110122, China

**Keywords:** animal experimental, biocompatibility, degradation performance, magnesium alloy, melatonin, osteoporotic fracture

## Abstract

Osteoporotic fractures are a very common bone disease that is difficult to completely cure. A large number of people worldwide suffer from pain caused by osteoporotic fractures every year, which can even cause disability and death. The compromised skeletal strength, lower density, trabecular microstructure, and bone-forming ability caused by osteoporotic fractures make them difficult to treat relative to normal fractures. An ideal scheme for osteoporotic fractures is to select internal fixation materials with matched mechanical and biological properties and carry anti-osteoporosis drugs on the plant to achieve bio-fixation and improve the condition of osteoporosis simultaneously. We designed a Mg-based MAO-MT-PLGA drug/ion delivery system (DDS) compatible with bone-like mechanical properties, degradation properties, and drug therapy. In this research, we evaluated the degradation behavior of Mg-based MAO-MT-PLGA DDS using immersion tests and electrochemical tests aided by SEM, EDS, XPS, XRD, and FT-IR. The DDS showed better corrosion resistance over Mg alloy and could release more Mg^2+^ due to the degradation of PLGA. According to cell viability and cell adhesion, the DDS showed better osteogenic characteristics over control group I (Mg alloy) and control group II (Mg-based MAO alloy), especially in the cells co-cultured with the leaching solution for 72 h, in which the DDS group increased to about 15% cell viability compared with group I (*p* < 0.05). The mRNA relative expressions, including ALP, collagen I, OCN, OPG, and Runx-2, as well as extracellular matrix calcium deposits of the DDS, are 1.5~2 times over control group I and control group II (*p* < 0.05), demonstrating a better ability to promote bone formation and inhibit bone resorption. After the DDS was implanted into the castrated rat model for one month, the trabeculae in the treatment group were significantly denser and stronger than those in the control group, with a difference of about 1.5 times in bone volume fraction, bone density, and the number of trabeculae, as well as the magnesium content in the bone tissue (*p* < 0.05). The above results demonstrated that the Mg-based MAO-MT-PLGA drug/ion delivery system is a potential treatment for osteoporotic fractures.

## 1. Introduction

Osteoporosis, the most common clinical bone disease in the middle aged and the elderly [1,2], is a systemic bone disease characterized by low bone mass, damage to bone microstructure, increased bone fragility, and susceptibility to fractures [3]. The biggest harm caused by osteoporosis is the osteoporotic fracture, and about nine million osteoporotic fractures occur worldwide each year [4,5,6]. Osteoporotic fracture is a sluggish and incremental process, accompanied by the thinning, breaking, and disappearance of the trabeculae of the cancellous bone in the early manifestations. The reduced number of trabeculae increases the load on the remaining trabeculae, resulting in significant fractures, and thus, the destruction of the bone structure. Further, one-third of the inner surface of the cortical bone gradually transforms into a structure similar to the cancellous bone, which makes the cortical bone thinner and results in a significant decrease in bone strength, including both elasticity and hardness [3,7]. The peculiar pathological features, including compromised skeletal strength, lower density, trabecular microstructure, and bone-forming ability caused by osteoporotic fractures make them difficult to treat relative to a normal fracture [8].

Implants commonly used to cure fractures are composed of stainless steel or titanium with high rigidity and a Young’s modulus far greater than that of healthy bones [9,10]. Poor bone quality contributes to decreased stability of internal fixation, often resulting in complications such as loosening, a need to extract internal fixation implants, or a failure of bone and grafts to fuse due to bone absorption [11]. Bone maintains its mineralization balance and its structural integrity through continuous reconstruction, and the mutual regulation between osteoblasts and osteoclasts is the basis for achieving the balance of bone formation and bone resorption [12]. The balance between bone formation and bone resorption is broken due to prolonged bed rest and immobilization, further aggravating the severity of the underlying disease [3], which is another extreme treatment difficulty posed by osteoporotic fractures.

Attributable to elastic modulus, similar to bones and degradable properties, Mg alloy is a potential material for treating osteoporotic fractures [9,10,13,14]. Our previous research results showed that the osteocytes lack magnesium while osteoporosis occurs [15]. Mg^2+^, which is beneficial for bone healing, is released with the degradation process proceeding. Numerous studies in vitro have paid attention to the effects of Mg^2+^ on the bone cells, in respect of enhancing proliferation and migration, as well as alkaline phosphatase (ALP) activity of human osteosarcoma MG-63 cells [16], increasing the viability and differentiation capacity of a human osteoblast cell line (hFOB1.19, ATCC) [17], improving cell proliferation of bone marrow-derived stromal cells (BMSC) [18], and promoting the expression of α2 and α3 integrins [19]. Furthermore, the fact that high concentrations of Mg^2+^ could modulate bone cell metabolism and bone cell function and inhibit the mineralization capacity of BMSCs has been proved [18,20]. Zhang et al. found magnesium when promoting calcitonin gene-related polypeptide-α (CGRP)-mediated osteogenic differentiation and developed a magnesium-containing intramedullary nail that promotes femoral fracture repair in castrated rats (osteoporosis model) [21].

Osteoporotic fractures are a combination of trauma and pathology, and local drug therapy can achieve an accelerated rebalance of bone resorption and bone reforming [22,23]. Bone remodeling is regulated by various drugs, such as BMP-2, BMP-7, bFGF, simvastatin, strontium ions, bisphosphonate, and melatonin (MT) [22]. It has been proved by numerous studies that MT is beneficial for osteoporosis treatment. Exogenous MT could activate the intracellular MEK/ERK1/2 signaling pathway by binding to the intracellular type II melatonin receptor, which promotes the osteogenic differentiation and calcification of MSCs [24]. Simultaneously, MT can inhibit the adipogenic differentiation that competes with osteogenic differentiation [25]. Unfortunately, there is disappointing physical behavior indicating that MT levels in the body decrease with age, especially in postmenopausal women [26]. Once the fracture occurs, systemic application of MT would lead to reduced efficiency or other safety concerns such as dizziness, anxiety, diarrhea, etc. W.P. Clafshenkel et al. designed a local delivery system (calcium-aluminate scaffolds) containing MT to enhance the osteoinductive and osteoconductive properties. Although bone regeneration was shown in the MT scaffold compared to the scaffold alone, the scaffold showed poor degradation in 6 months [27].

An ideal scheme for osteoporotic fracture is to select internal fixation materials with matched mechanical, degradable, and biological properties, and carry anti-osteoporosis drugs on the plant to achieve bio-fixation and improve the condition of osteoporosis simultaneously. It is a prospective treatment method for osteoporotic fracture, which combines the Mg alloy implant with MT.

Typically, MT is encapsulated in microspheres for research and use [28,29]. However, to our knowledge, despite the safety, targeting, and other therapeutic features of microspheres, microspheres are mostly administered by injection, which will increase the treatment burden for fractured patients. Meanwhile, microspheres are not easy to prepare directly on Mg alloys due to the lack of intermediate grafts. Even if the microspheres are combined by the coating, it is easily lost due to operations such as shaping and friction during the surgery, making it difficult to achieve the designed therapeutic effect.

Based on the above, Mg-based MAO alloy, prepared via micro-arc oxidation of Mg alloy, was selected as a drug carrier for MT owing to the excellent adhesion of the MAO film. The MT was combined with a silane coupling agent, and the outermost surface was covered with a layer of PLGA to accelerate the release of MT and Mg^2+^. An Mg-based MAO-MT-PLGA drug/ion delivery system (DDS) compatible with bone-like mechanical property, degradation property, and drug therapy is designed for the local treatment of osteoporotic fractures. To our knowledge, no researchers have used Mg alloys as both an implant and a melatonin-loaded carrier for the treatment of osteoporosis. As a type of newly designed DDS, it is essential to ensure that there is no potential toxicity during the degradation of the DDS. Since the Mg alloy substrate is a material that can be degraded in vivo, the whole degradation process, including the release of MT and Mg^2+^, should be clarified compared to the Mg alloy and the Mg-based MAO alloy to understand the mechanism of action of the DDS. It is also necessary to prove the reliability of the DDS through animal experiments. Hence, in this article, degradation behaviors, biocompatibility, and osteogenic performance were evaluated in vitro, and animal experiments for treating castrated rats (osteoporosis model) were carried out in vivo, which will prove that the Mg-based MAO-MT-PLGA DDS is full of potential to treat osteoporotic fractures.

## 2. Materials and Methods

### 2.1. Samples Preparation

A self-developed Mg-2Zn-0.5Nd-0.5Zr alloy (alloys were manufactured by Changchun Institute of Applied Chemistry Chinese Academy of Sciences, Changchun, China) with high mechanical properties and a low degradation rate was chosen as the control group I, Mg alloy group. Mg-based MAO alloy, as control group II, was prepared via micro-arc oxidation with the specific parameters as follows: the power-supply rated power was 2000 V, and the operating voltage, frequency, and duty cycle were 400 V, 1000 Hz, and 6 min, respectively. The electrolyte was composed of 8 g/L KF, 4 g/L (NaPO_3_)_6_ and 1.2 g/L Ca(OH)_2_ (the above drugs were purchased at Sinopharm Chemical Reagent Co., Ltd., Shanghai, China).

Silane-Coupled MT/poly (lactic-co-glycolic acid) composite coating (SiH_4_-MT/PLGA composite coating) was fabricated on the surface of the MAO film. The composition of the silane coupling agent was silane: alcohol: purified water at 1:1:10, where 3 gradients of MT (25 nM, 50 nM, 75 nM) were dissolved as the SiH_4_-MT solution. The Mg-based MAO alloy samples were immersed in the SiH_4_-MT solution for 10 min and pulled 5 times at a speed of 1 cm/min to form a SiH_4_-MT layer. Additionally, samples with a SiH_4_-MT layer were then immersed in the chloroform solution of 5 g/L PLGA (LA:GA = 75:25, 100 thousand molecular weight) for 5 min and pulled 3 times at a speed of 1 cm/min, forming a SiH_4_-MT/PLGA composite coating. The processed whole sample, as the experimental group, was defined as the Mg-based MAO-MT-PLGA drug/ion delivery system (DDS). (The PLGA was purchased at Jinan Daigang Biomaterial Co., Jinan, China; the SiH_4_ was purchased at Maya Reagent, Jiaxing, China; and the MT was purchased at Beijing Solarbio Science & Technology Co., Beijing, China).

### 2.2. Biodegradation Behavior

In this section, the immersion test and electrochemical test were carried out at 37 °C. Different groups of samples were immersed in Hank’s solution for 28 days, with an immersion ratio of 1.25 cm^2^/mL. The pH value of the solution was monitored at the same time of day. To keep the solution fresh, the solution was replaced daily during the entire cycle. Additionally, the corrosion morphologies and corrosion product composition of samples that were immersed for 7 days, 14 days, and 28 days were investigated by SEM (FEI Inspect F50, Hillsboro, OR, USA), XRD (Rigaku Dmax 2500 PC, Tokyo, Japan), FT-IR (Nicolet 6700, Glendale, WI, USA) and XPS (ESCALAB 250, Richmond, VA, USA) [30]. The degradation behavior of samples was evaluated via an electrochemical test, including open-circuit potential (OCP), electrochemical impedance spectroscopy (EIS), and potentiodynamic polarization through the electrochemical workstation (Gamry Reference 600+, Warminster, PA, USA) [31]. The three-electrode system was used during the whole test, with the calomel electrode as the reference electrode, the platinum electrode as the counter electrode, and the sample as the working electrode. OCP was measured for half an hour, and then EIS was measured with the frequency of 10^5^~0.01 Hz. Finally, the potentiodynamic polarization test was performed at a scan rate of 0.5 mV/s from −0.3 V to 0.5 V in reference to OCP.

### 2.3. Biological Properties In Vitro

#### 2.3.1. Cell Culture

The biological experiments used adult osteoblasts (hFOB1.19) to study the interaction of materials with cells in vitro. Cells were first placed in an α-MEM medium (Hyclone, Logan, UT, USA) containing 10% fetal bovine serum (BI, Minneapolis, MN, USA) and 100 U/mL dual antibodies (penicillin and streptomycin) in an incubator (37 °C, 95% Humidity, and 5% CO_2_). After 2–3 days, when the cells grew to cover about 80% of the culture dish, passage treatment was carried out. The culture dish was rinsed twice with phosphate-buffered solution (PBS), and the cells were digested with EDTA-containing trypsin until they contracted. The culture dish was shaken gently until it was observed that the cells were about to fall off the culture flask, and then the same volume of complete medium was added to terminate the digestion. The cell suspension concentration was counted using a hemocytometer prior to the experiment.

#### 2.3.2. Preparation of Leaching Solution

After ultrasonic cleaning of the Mg alloy, Mg-based MAO alloy, and Mg-based MAO-MT-PLGA DDS samples in absolute ethanol, they were quickly rinsed and dried with deionized water, sterilized using UV lamp irradiation, and sterilized via irradiation on the front and back sides for 30 min. Then, they were added to the above-mentioned α-MEM medium at a ratio of 1.25 cm^2^/mL, and the leaching solution was obtained via leaching in an incubator for a certain period.

#### 2.3.3. Biocompatibility

For the CCK-8 assay measurement of cell viability, the cells (hFOB1.19) were treated as described below. Before treatment, the cells were plated on 96-well plates for 24 h. After a series of leaching solution co-culture for 24 h, 48 h, and 72 h, each well was added to CCK-8 solution and incubated for 1 h. Absorbance was determined with a microplate reader (Multiskan SkyHigh, Waltham, MA, USA) at 450 nm. The results were performed with percentages, while the untreated cells were taken into 100% [32].

For the cytoskeleton staining assay, the cells were plated on 12-well plates for 24 h for cell sticking and then were cultured in a leaching solution at 37 °C under a CO_2_ atmosphere for 7 days. After discarding the leaching solution in the wells, cells were fixed with 4% paraformaldehyde for 30 min, and 0.5% Triton X-100 permeabilizer was permeabilized for 5 min at room temperature. By adding DAPI to stain the nuclei, the images were observed under a fluorescence microscope (Olympus IX71, Tokyo, Japan).

### 2.4. Osteogenic Performance In Vitro

Total RNA was isolated using MiNiBEST universal RNA Extraction Kit (Takara, Kusatsu, Japan) according to the manufacturer’s instructions, and reverse transcribed using Primescript RT Master Mix (Takara, Kusatsu, Japan). Real-time PCR was performed on a Lightcycler 480 real-time PCR system (Roche, Pleasanton, CA, USA) with a SYBR premix ex Taq II Kit (Takara, Kusatsu, Japan). Table 1 lists the methods used to amplify alkaline phosphatase (ALP), collagen I, OCN, OPG, and Runx-2, as well as β-Actin, which was used as an internal control. The experimental data were processed using the 2^−ΔΔCt^ method, as follows: ΔΔCt = (Ct target − Ct internal control) experiment group − (Ct target − Ct internal control) control group. Each experiment was repeated three times.

After culturing with an osteogenic medium for 14 days, the cells were washed and fed routinely, and the extracellular matrix calcium deposits were measured by staining the cells with 40 mM alizarin red S solution (Sigma-Aldrich, Taufkirchen, Germany). The cells and nodule formation were captured using phase-contrast microscopy (Nikon, Tokyo, Japan). The density was analyzed via Image-Pro Plus 6.0.

### 2.5. Animal Experimental

In order to verify the effectiveness of the Mg-based MAO-MT-PLGA DDS in the treatment of osteoporosis, the samples of the Mg-based MAO-MT-PLGA DDS were planted on the castrated rat (osteoporosis model) proximal tibia for one month, where rats were divided into a treatment group and osteoporosis bone group as control [33]. The treatment effect is mainly determined by bone trabecula density. Proximal tibia after treatment with the Mg-based MAO-MT-PLGA DDS was reconstructed by the Micro-CT (SKYSCAN 1276, Germany). The tibia specimens of rats in each group were taken and placed in 10% neutral formaldehyde buffer fixative solution (pH = 7.2~7.4, 4 °C) for 24–48 h, and then treated with 10% EDTA-2 Na buffer solution (pH = 7.4, 4 °C) for decalcification, and the decalcification solution was changed every 5–7 days; after the decalcification was complete, the cells were dehydrated with graded ethanol, transparent in xylene, embedded in longitudinal paraffin, serially sectioned at 5 μm, and stained with conventional HE. Mg^2+^ content in bone tissue was detected by the magnesium ion fluorescent probe and the fluorescence intensities were quantified using the ImageJ v2021.8.0 software.

### 2.6. Statistical Analysis

The quantitative data were the mean ± standard deviation (SD). All data were analyzed using the software program GraphPad Prism 6.02. Comparisons among multiple groups were performed using a one-way analysis of variance. Pairwise comparisons were performed using the *t*-test. *p* < 0.05 indicated that the observed difference was significant.

## 3. Results

### 3.1. Surface Morphology

The surface morphology of the Mg-based Mao alloy and Mg-based MAO-MT-PLGA DDS was investigated via SEM. Figure 1 shows the SEM images of surface morphology of the Mg-based Mao alloy and the Mg-based MAO-MT-PLGA DDS. As Figure 1a,b show, continuous molten oxide particles existed on the surface of the Mg alloy, accompanied by lots of tiny and uniform pores, which were formed by the oxygen bubbles in the coating growth process and the thermal stress as a result of the rapid solidification of the molten oxide in the relatively cooling electrolyte. Figure 1c shows the element content of the MAO film of the Mg-based Mao alloy. The surface morphology of the Mg-based MAO-MT-PLGA DDS is shown in Figure 1d,e. There is no discernible difference between the Mg-based MAO alloy and Mg-based MAO-MT-PLGA DDS in terms of morphology. In terms of element analysis, it can be seen that MT adheres to the surface of the MAO film in Figure 1f. Figure 1e attaches the surface morphology of MT. XRD results of the Mg-based Mao alloy and Mg-based MAO-MT-PLGA DDS are shown in Figure 2a, limited by the requirement of content (>5%) in the XRD test; MT is not detected. MAO film is composed of phase MgO, Ca_3_(PO_4_)_2_, Mg_3_(PO_4_)_2_, Mg(H_2_PO_4_)_2_, and CaHPO_4_. Figure 2b shows the FT-IR pattern of the Mg-based Mao alloy and Mg-based MAO-MT-PLGA DDS. The infrared absorption peak at 1515 cm^−1^ is caused by the deformation vibration of the OH bond, but the absorption peak at this place is obviously broadened in the Mg-based MAO-MT-PLGA DDS, which may be caused by the overlapping of multiple peaks. The positions of C=O double bonds in MT and C=C double bonds in the benzene ring exist in this region, most likely resulting in a broadening of the absorption peak there. The infrared characteristic absorption peak at 1004 cm^−1^ may be caused by the stretching vibration of a small amount of P=O double bond, while there is a shoulder near the absorption peak of the Mg-based MAO-MT-PLGA DDS group, which is contributed by the stretching vibration of the C-O bond in MT.

### 3.2. Degradation Behavior

The immersion test reflects the degradation process of the Mg alloy, Mg-based MAO alloy, and Mg-based MAO-MT-PLGA DDS in a certain period in vitro. Figure 3a shows the pH value of the Mg alloy, Mg-based MAO alloy, and Mg-based MAO-MT-PLGA DDS after immersion in Hank’s solution for 28 days. With the prolongation of immersion time, the changing trend of the pH value among the three groups was similar, showing a rapid increase in the first two days, and then a gradual decrease. The pH value indirectly reflects the amount of Mg degradation in each Hank’s solution exchange cycle, meaning that the degradation rate of Mg alloy is higher than that of Mg-based MAO alloy and Mg-based MAO-MT-PLGA DDS. The performance of Mg-based MAO-MT-PLGA DDS is not as resistant as the Mg-based MAO alloy due to accelerated degradation after PLGA hydrolysis. Floes are observed in Hank’s solution for Mg-based MAO-MT-PLGA DDS after immersion for two days, which confirms the surge of magnesium ions on the third day (Figure 3b).

Figure 4a compares the surface macro morphologies of the samples before pickling after immersion for 28 days. A coating composed of degradation products forms on the Mg alloy after immersion for 28 days, while only part of the MAO film of Mg-based MAO alloy and Mg-based MAO-MT-PLGA DDS is damaged, and the whole is still protected by the MAO film. The surface macro morphologies of the samples after pickling after immersion are shown in Figure 4b, where the surface of Mg alloy has filiform corrosion and serious pitting corrosion, while the surfaces of the Mg-based MAO alloy and Mg-based MAO-MT-PLGA DDS after pickling only exhibit filiform corrosion characteristics, which is the commonality of the corrosion behavior of Mg alloy coatings [34].

The surface micromorphology after immersion for 7 days, 14 days, and 28 days is shown in Figure 5. The degradation products cover the surface of magnesium alloy, the surface degradation product layer thickens, and cracks appear with the extension of immersion time. In contrast, the corrosion morphology on the surface of the Mg-based MAO alloy and Mg-based MAO-MT-PLGA DDS is similar. After immersion for 28 days, small pores on the Mg-based MAO alloy can still be seen, accompanied by fewer corrosion products. Compared with the Mg-based MAO alloy, Mg-based MAO-MT-PLGA DDS shows a more violent corrosion surface after 14 days and 28 days, displaying that corrosion occurs from the pores of MAO film and starts to expand, which is speculated to be caused by the degradation of PLGA. The cross-section images of the Mg alloy, Mg-based MAO alloy, and Mg-based MAO-MT-PLGA DDS after immersion for 28 days are shown in Figure 6. The surfaces of different samples are covered by film layers. Despite the thickness order of Mg-based MAO alloy > Mg-based MAO-MT-PLGA DDS > Mg alloy, corrosion resistance is consistent with the trend of pH change, which can be explained by the film composition. The main component of the Mg-based MAO sample surface film is MgO formed during the discharge remelting process, which has a better protective effect than the loose film layer composed of degradation products on the surface of the Mg alloy. Meanwhile, the thinner cross-section thickness of Mg-based MAO-MT-PLGA DDS than that of Mg-based MAO alloy manifests the hydrolysis of PLGA again. Deep pits in the cross-section are caused by the excessive brittleness of the MAO film during the polishing process.

The content changes of main elements from the surface to the interior of the degradation products are analyzed by XPS. As shown in Figure 7a, Mg, O, Ca, P, K, Cl, and C elements are tested from the surface to the inner layer. Mg, O, Ca, and P are the major elements in the products, and each content increases with the increase in etch time. C in the surface layer mainly comes from CO_2_ in the air and gradually decreases inward. By analyzing FT-IR results (Figure 7b), PO_4_^3−^ is found in the degradation products of the Mg alloy, Mg-based MAO alloy, and Mg-based MAO-MT-PLGA DDS after immersion for 28 days, which is mainly owing to the fact that Hank’s solution is a phosphate-buffered solution. Combined with the analysis results of XRD (Figure 7c), it can be found that the main components of degradation products are Ca_3_(PO_4_)_2_ and Mg_3_(PO_4_)_2_.

Potentiodynamic polarizatiotention and EIS curves reflect the short-term degradation process of the Mg alloy, Mg-based MAO alloy, and Mg-based MAO-MT-PLGA DDS in Hank’s solution, presented in Figure 8. The results of Tafel fitting of potentiodynamic polarizatiotention curves (Figure 8a) are listed in Table 2. The corrosion current density (I_corr_) is a parameter closely related to the corrosion resistance of the sample. The smaller the corrosion current density, the better the corrosion resistance. The corrosion rate of the Mg alloy, Mg-based MAO alloy, and Mg-based MAO-MT-PLGA DDS samples can be calculated with reference to the corrosion current density, and the overall result is consistent with the immersion test. Nyquist plots in Figure 8b can be divided into three parts, namely capacitive reactance arcs in the high and mid-frequency regions and inductive reactance arcs in the low-frequency region. The difference in the EIS spectrum is mainly in the radius of the capacitive reactance arc, and the radius of the capacitive reactance spectrum is positively correlated with the corrosion resistance. That is to say, the EIS results show that the Mg-based MAO-MT-PLGA DDS has better corrosion resistance than the Mg-based MAO alloy and Mg alloy. Inconsistencies with the results of the immersion experiments will be discussed intensively in the Section 4. Values of equivalent elements in impedance models of different samples are listed in Table 3.

### 3.3. Biocompatibility

The cell viability of Mg alloy, Mg-based-MAO alloy, and Mg-based MAO-MT-PLGA DDS with three concentration gradients was evaluated after a series of treatments (24 h, 48 h, 72 h). There was no obvious difference between all groups 24 h later. However, group differences began to be expressed at 48 h and was further differentiated at 72 h, with the Mg-based MAO-MT-PLGA DDS having higher cell viability than Mg-based MAO alloy and Mg alloy, and the latter performing comparably. Among the three concentrations of the Mg-based MAO-MT-PLGA DDS, the DDS with 50 nM MT showed the best biological activity, so all Mg-based MAO-MT-PLGA DDS samples used in this article were performed with this parameter (Figure 9a).

The spreading of the cytoskeleton is an important indicator of cell adhesion and morphology. By observing the cells co-cultured with the leaching solution for 24 h, the three groups of cells have good spreading morphology and significant actin expression. Cell adhesion behavior is not affected after surface modification (Figure 9b).

### 3.4. Osteogenic Performance In Vitro

Osteoblasts can secrete ALP, synthesize extracellular matrices such as collagen I, OCN, and OPG, and further mineralize to form bone tissue. These characteristics are used as criteria for identifying bone marrow mesenchymal stem cells differentiated into osteoblasts. In addition, ALP is an enzyme necessary for bone formation and an early marker of osteoblast differentiation and functional maturation [35,36]. Runx-2 is the key regulator to regulate the differentiation of osteoblasts and osteoclasts and promote bone formation. By regulating the expression of osteoblast-specific extracellular matrix protein gene and osteoblast cycle, Runx-2 participates in the differentiation process of osteoblasts, promotes bone formation, and inhibits bone resorption [37]. Using PCR tests, the mRNA relative expression of ALP, collagen I, OCN, OPG, and Runx-2 is in progress, and the Mg-based MAO-MT-PLGA DDS demonstrates a better ability to promote bone formation and inhibit bone resorption beyond Mg alloy and the Mg-based MAO alloy (Figure 10a). The performance between the Mg alloy and the Mg-based MAO alloy is almost identical, except in the OPG test, where the Mg-based MAO alloy shows the lower relative expression.

For measurement of bone nodule formation after osteogenic differentiation, detection of the extracellular matrix calcium deposits was determined using ARS staining (Figure 10b). The results showed that the relative calcium deposition in the Mg-based MAO-MT-PLGA was much higher than that in the Mg alloy and the Mg-based MAO alloy.

### 3.5. Animal Experimental

Trabeculae is the general reference for the analysis of osteoporosis. The images of the proximal tibia of the treatment group rats and osteoporosis group rats reconstructed by the Micro-CT are shown in Figure 11. It can be seen that the tibial trabeculae of the Mg-based MAO-MT-PLGA DDS treatment group were dense and intertwined into a mesh, while in the control group, the trabeculae number was significantly reduced, and the trabeculae became thinner and more fractured. By calculation, the use of the Mg-based MAO-MT-PLGA DDS to treat osteoporosis can increase bone volume fraction and bone mineral density by about 1.5 times compared with the control group. Furthermore, trabecular numbers and trabecular thickness are calculated, which also shows that the treatment group was much higher than the control group. HE staining can visually express the distribution of trabeculae. As shown in Figure 11, the distribution of trabeculae in the treatment group was dense and compact per unit area, while the number of trabeculae in the control group was significantly less and the distribution was sparse. The semi-quantitative analysis of trabeculae shows that the trabecular number of the treatment group was over 1.5 times that of the control group. The above proves that the Mg-based MAO-MT-PLGA DDS has a good therapeutic effect on osteoporosis.

## 4. Discussion

### 4.1. Degradation Process and MT/Mg^2+^ Release

The degradation process is mainly the oxidation reaction of magnesium with the participation of H_2_O, which can be described as the following equations.
(1)$$\mathrm{Mg}+2H_{2}O\to {\mathrm{Mg}}^{2+}+H_{2}+2{\mathrm{OH}}^{-}$$

The Equation (1) indicates that H_2_ and OH^−^ are produced in the reaction at a 1:2 molar ratio. The generation of H_2_ and OH^−^ causes severe safety concerns in the early stage of Mg alloy implantation, such as the appearance of air pockets and localized osteolysis of high alkalinity.
(2)$${\mathrm{Mg}}^{2+}+2{\mathrm{OH}}^{-}\to \mathrm{Mg}{\left(\mathrm{OH}\right)}_{2}$$

As the Mg^2+^ increases, the hydrolysis reaction occurs, as shown in Equation (2). The resulting Mg(OH)_2_ is a preliminary protective film to reduce the degradation rate. In addition, H_2_PO_4_^−^ and HPO_4_^2−^ combine a buffer pair in Hank’s solution to maintain the pH value. In the alkaline solution, they react with OH^−^ to try to maintain pH. Equations (3) and (4) describe these procedures.
(3)$$H_{2}{\left({\mathrm{PO}}_{4}\right)}^{-}+{\mathrm{OH}}^{-}\to {\mathrm{HPO}}_{4}{}^{2-}+H_{2}O$$
(4)$${\mathrm{HPO}}_{4}{}^{2-}+{\mathrm{OH}}^{-}\to {\mathrm{PO}}_{4}{}^{3-}+H_{2}O$$
(5)$$3{\mathrm{Mg}}^{2+}+2{\mathrm{HPO}}_{4}{}^{2-}+2{\mathrm{OH}}^{-}\to {\mathrm{Mg}}_{3}{\left({\mathrm{PO}}_{4}\right)}_{2}+2H_{2}O$$
(6)$$3{\mathrm{Mg}}^{2+}+2{\mathrm{PO}}_{4}{}^{3-}\to {\mathrm{Mg}}_{3}{\left({\mathrm{PO}}_{4}\right)}_{2}$$

Equations (5) and (6) are the major reactions to form degradation products after immersion in Hank’s solution for 28 days. As the FT-IR, XRD and XPS results show, ${\mathrm{Mg}}_{3}{\left({\mathrm{PO}}_{4}\right)}_{2}$ is major component of degradation products, playing an important role to resist corrosion. Similarly, Ca^2^^+^ in solution also reacts, as shown in Equations (7) and (8).
(7)$$3{\mathrm{Ca}}^{2+}+2{\mathrm{HPO}}_{4}{}^{2-}+2{\mathrm{OH}}^{-}\to {\mathrm{Ca}}_{3}{\left({\mathrm{PO}}_{4}\right)}_{2}+2H_{2}O$$
(8)$$3{\mathrm{Ca}}^{2+}+2{\mathrm{PO}}_{4}{}^{3-}\to {\mathrm{Ca}}_{3}{\left({\mathrm{PO}}_{4}\right)}_{2}\;$$

According to the changes of pH (Figure 3), Mg alloy with coatings is required to resist the faster degradation rate in the early stages. The difference in the degradation process and ion release between Mg alloy and Mg-based MAO alloy can refer to the following Figure 12a,b.

Pores on the surface of Mg-based MAO alloy are the unique channel for the communication between the Mg alloy substrate and the body fluid. Hence, less ion exchange occurs in the Mg-based MAO alloy per unit time, which means lower corrosion behavior occurs. On the one hand, MAO film can well protect the Mg alloy matrix, but the insufficient release of Mg^2+^ also has an impact on the osteoporosis treatment, which can be found in the detection of OPG by PCR [38]. Once osteogenesis and osteoclasts are out of balance, osteoporosis will worsen and fractures will be more difficult to heal [12]. It is not advisable to treat osteoporotic fractures with overprotected magnesium alloys.

PLGA, randomly polymerized from lactic acid and glycolic acid, is a biodegradable polymer with good biocompatibility, non-toxicity, and good film-forming properties [39]. Lactic acid and glycolic acid, the degradation products of PLGA, will hydrolyze with hydroxyl groups generated during the degradation of Mg alloys, accelerating the release of magnesium ions, which is proved by the following results. The pores on the Mg-based MAO-MT-PLGA DDS surface after immersion in Hank’s solution for 14 days and 21 days open out and are covered by the degradation products. In contrast, the pores on the Mg-based MAO alloy surface have only minor enlargement and no obvious degradation product coverage. The cross-section of the Mg-based MAO-MT-PLGA DDS is thinner than that of the Mg-based MAO alloy. During the preparation of the leachate, the Hank’s solution of the Mg-based MAO-MT-PLGA DDS began to appear with tiny white flocs on the second day. This was a kind of preliminary degraded PLGA. This is also evidenced by the amount of ion accumulation in standard Hank’s solution (Figure 3b). It is speculated that the degradation of the PLGA film is also the main reason for the difference between the EIS results and the immersion results. EIS can be described as a quasi-steady-state method. During its operation, a small-amplitude sinusoidal potential signal is applied to perturb the system, and anodic and cathodic processes alternate on the electrodes. Even if the perturbation signal acts on the electrode for a long time, it will not lead to the cumulative development of the polarization phenomenon and the cumulative change of the electrode surface state. Attributable to the protective effect of PLGA in a short time, the EIS results show the Mg-based MAO-MT-PLGA has better corrosion resistance. The degradation process and drug/ion release of the Mg-based MAO-MT-PLGA DDS can be described in Figure 12c.

Our previous study about the Mg alloy used in this research shows the Mg-2Zn-0.5Nd-0.5Nd owns a proper degradation rate, either in vitro or in vivo. The degradation rate of the Mg-2Zn-0.5Nd-0.5Zr alloy extruded bars in the Hank’s solution tested by Chen is about 0.6 mm/year [40]. Jin implanted the pure Mg and Mg-2Zn-0.5Nd-0.5Nd alloy in the mandible of New Zealand rabbits simultaneously, and found that pure Mg caused localized osteolysis, while Mg-2Zn-0.5Nd-0.5Nd did not. Based on previous studies, it can be concluded that the Mg-based MAO-MT-PLGA DDS is suitable for implanting in terms of degradation behavior.

### 4.2. Biocompatibility

In most cases, MAO films and their composites are applied on the surface of magnesium alloys to increase their corrosion resistance. Numerous studies have been conducted and have proven the corrosion resistance effectiveness of MAO film and its composites [41,42]. However, it is inadequate for MAO film to improve the cytocompatibility because cells can hardly accommodate themselves on the material according to the previous research [43], which is further confirmed by the results of cytocompatibility tests between Mg alloy and Mg-based MAO alloy in this study. It is attributable to the cause that the surface of magnesium-based micro-arc oxidation is dominated by MgO ceramic phase. The binding of MT, the osteogenic activity factor, on the MAO film stimulated the cell biological activity of the Mg-based MAO-MT-PLGA DDS.

### 4.3. Cells and Osteoporotic Bone Response to the Mg-Based MAO-MT-PLGA DDS

As previously mentioned, the intracellular type II melatonin receptor binds to exogenous melatonin to promote osteogenic differentiation and calcification of MSCs by activating the intracellular MEK/ERK1/2 signaling pathway [24], which explains the superior gene expression of the Mg-based MAO-MT-PLGA DDS in PCR assays. It has been found that magnesium deficiency affects the expression level of OPG [38]. Mg^2+^ originating from the Mg-based MAO alloy releases less than Mg alloy per unit time owing to the film protection, which may be the reason for affecting OPG mRNA relative expression between Mg alloy and Mg-based MAO alloy. MT can promote the proliferative activity of MSCs adhered to the extracellular matrix scaffold and promote the generation of extracellular matrix calcium deposits [44]. Despite the hFOB1.19 cell used in this research, similar results were shown in terms of cell viability and calcium deposition.

The castrated rat model, a living model replicated by removing the bilateral ovaries of rats to artificially induce ovarian function defects, is a classic model for the study of postmenopausal osteoporosis (PMOP). The characteristics and results of changes in bone metabolism in this model are similar to changes in PMOP: increased bone turnover, greater bone resorption than bone formation; initial rapid bone loss phase, followed by slow bone loss phase, with more loss of cancellous bone than cortical bone; decreased intestinal absorption of calcium, and increased urinary calcium excretion. Interestingly, compared with human osteoporosis, the castrated rat model has a similar bone tissue response to estrogen, fluoride, bisphosphonate, PTH, calcitonin, and physical exercise [45].

In this study, judging by the differences in the changes of bone trabecular, using the Mg-based MAO-MT-PLGA DDS to treat OP has achieved gratifying results, where the reason for the positive effect lies in melatonin and magnesium ions. Previous studies have shown that once the competitive balance between bone and fat is broken, osteoporosis may occur, which is characterized by loss of bone mass and a linear increase in the number of cells in the medullary cavity, and fat replaces bone trabecular [46]. MT has been demonstrated to regulate the osteogenic and adipogenic differentiation of stem cells. Zhang et al. reported that MT inhibited adipogenic differentiation of stem cells and promoted osteogenic differentiation in a concentration-dependent manner [25]. Additionally, he also found that MT down-regulated markers of terminal adipocyte differentiation, such as leptin, lipoprotein lipase, adiponectin, and adipocyte proteins, while up-regulating markers of osteoblast differentiation, such as ALP, osteopontin, and osteocalcin [25]. Radio et al. studied and demonstrated that melatonin intervention increased the degree of coupling of MT2 receptors to Giα2, β-arrestin-1, β-arrestin-2, MEK1/2, and ERK1/2, and there was no association between MT2 receptors and EGFRs, and no association between MT1 receptors and Giα2, β-arrestin-1, β-arrestin-2, MEK1/2, ERK1/2, or EGFRs. Meanwhile, the melatonin receptor antagonist Luzindole can block the formation of these complexes, suggesting that melatonin-promoted osteogenic differentiation may be mediated through the formation of the MT2R/Gi/b-arrestin/MEK/ERK1/2 complex [44,47]. However, except for the MT1 and MT2 membrane receptors that have been extensively studied, other MT binding sites, such as nuclear receptors ROR and RZR isoforms derived from the retinoic acid receptor superfamily, quinone reductase-2, calmodulin, calreticulin, and mitochondrial binding sites are found as well [25]. It is difficult to clarify the regulatory mechanism of melatonin on the osteogenic differentiation of stem cells.

The osteogenic or adipogenic differentiation of stem cells is regulated by multiple transcription factors, which act as switches to regulate whether stem cells differentiate osteogenically or adipogenically. In adipogenic differentiation, key transcription factors C/EBP β and PPARγ are inhibited by MT [48]. Luzindole could not antagonize the effect of MT on adipogenic inhibition, which means that MT mediates osteogenic and adipogenic differentiation through different signaling pathways [49]. Melatonin is considered a ligand of the orphan nuclear receptor ROR/RZR family [50]. Ohoka et al. found that the orphan nuclear receptor RORα inhibits adipocyte differentiation by downregulating C/EBPβ activity and perilipin gene expression. The study also found that the mRNA expression of the RORα receptor could be detected during the adipogenic process of human mesenchymal stem cells. If RORα receptor expression is activated, adipogenesis can be inhibited. If the expression of the RORα receptor is inhibited by RNAi transfection, it will promote the adipogenic differentiation of stem cells. Therefore, the RORα orphan nuclear receptor may play a mediating role in the regulation of adipogenesis by melatonin [51].

The released Mg^2+^ plays a significant role in treating osteoporotic bone. Figure 11c shows the magnesium content in the tissues of the treated and osteoporosis groups, and the former is much higher than the latter, which means Mg^2+^ released by the degradation process can be absorbed by osteoporotic bone efficiently. Our previous study found that magnesium deficiency will induce apoptosis in type 2 diabetic osteoporosis adjusted by NIPA2 [15]. Abnormal apoptosis of osteoblasts can disrupt the balance between adipogenic differentiation and osteogenic differentiation of stem cells, thereby accelerating the symptoms of osteoporosis. High concentrations of Mg^2+^ can regulate osteocyte metabolism and osteocyte function, and the Wnt/β-catenin anti calcification pathway and the magnesium transporter SLC41A1 have been shown to be involved in magnesium-mediated signaling in BMSCs [20]. Therefore, in terms of the treatment of osteoporotic fractures, on the one hand, Mg alloys with coatings should have a sufficient protective effect and on the other hand, should release sufficient Mg^2+^.

## 5. Limitations

In conclusion, the Mg-based MAO-MT-PLGA DDS accomplishes its goal of treating osteoporosis by releasing Mg^2+^ and MT. Although the Mg-based MAO-MT-PLGA DDS shows good potential for the treatment of osteoporotic fractures, there are a number of issues that deserve further exploration. The highest tensile strength of the degradable magnesium alloys that have been reported is close to 400 MPa [52], and we have also carried out a series of optimization studies on the strength of Mg-2Zn-0.5Nd-0.5Zr alloys, which can reach this level. However, osteoporotic fractures mostly occur in load-bearing parts, such as the spine, hip, and wrist, requiring magnesium alloys to have higher mechanical properties. So far, no medical magnesium alloy can be adapted. It is an urgent and significant problem to further improve the mechanical properties of biological Mg alloys without sacrificing biocompatibility, which means that no fortification with toxic elements such as aluminum, nor excess rare earth elements can be used. Porous materials [53] and bioactive glass [54] are also directions that can be tried in the future.

During the early stages of the entire bone regeneration process, especially during the first 14 days, accelerated stem cell and vascular-migration-increased inflammatory responses, as well as immune modulation, could promote bone regeneration [23]. Therefore, an ideal drug delivery system releases MT within 14 days. In this article, PLGA plays a role in accelerating the release of Mg^2+^ via its hydrolysis. Achieving controlled release by changing the molecular weight of PLGA, film thickness, or blending PLGA with other polymers is not presented in this paper, which is also the focus of future research. Furthermore, in the process of treating osteoporosis, the mechanism of the interaction between MT and Mg^2+^ is still unclear. Simultaneously, this study only evaluates the effectiveness of the Mg-based MAO-MT-PLGA DDS for the osteoporosis model. A bigger animal experiment is necessary and more persuasive [45]. Next, we will evaluate the effectiveness of the Mg-based MAO-MT-PLGA DDS by implanting bone plates into a New Zealand White Rabbit Fracture Model.

## 6. Conclusions

This study designed a new type of Mg-based MAO-MT-PLGA DDS for treating osteoporotic fractures and evaluated its biocompatibility and degradation properties. We found that the cell viability, mRNA relative express, and calcium deposition of the Mg-based MAO-MT-PLGA DDS are far superior to Mg alloy or Mg-based MAO alloy in terms of the cellular level. PLGA appropriately accelerated the Mg^2+^ release, and MAO coating protects Mg substrate from rapid corrosion to some extent. The Mg-based MAO-MT-PLGA DDS was further implanted into the proximal tibia of castrated rats, and the trabecular angle osteoporosis group was significantly improved after one month, and more magnesium content was detected in the tissue, which demonstrates the effectiveness of the Mg-based MAO-MT-PLGA DDS in the treatment of osteoporotic fractures at the animal level. Although the Mg-based MAO-MT-PLGA DDS has been shown to be very promising for the treatment of osteoporotic fractures, the mechanical properties of Mg alloy, the controlled release of PLGA, and the mechanisms of action of MT and Mg^2+^, and further bigger animal experiments remain to be studied.

## Figures and Tables

**Figure 1 pharmaceutics-14-01481-f001:**
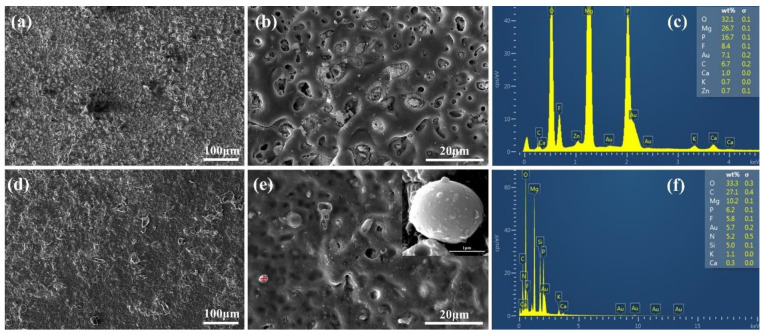
The surface morphology of Mg-based MAO alloy (**a**,**b**), the surface morphology of Mg-based MAO alloy (**d**,**e**), “+” in the (**e**) shows the MT particle attached to the MAO film surface, the element content of the MAO film of the Mg-based Mao alloy (**c**), the element content on the MT particle surface (**f**).

**Figure 2 pharmaceutics-14-01481-f002:**
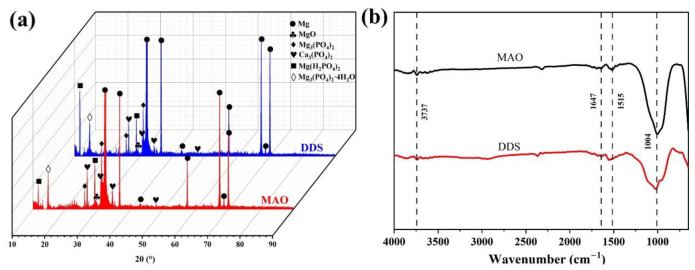
XRD patterns (**a**) and FT-IR (**b**) patterns of Mg-based MAO alloy and Mg-based MAO-MT-PLGA DDS.

**Figure 3 pharmaceutics-14-01481-f003:**
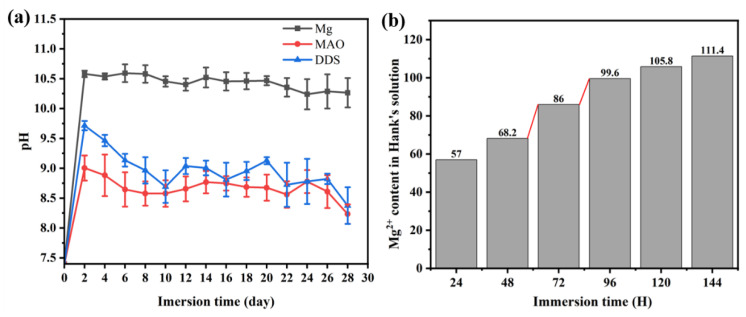
Change of pH value of Mg alloy, Mg-based MAO alloy, and Mg-based MAO-MT-PLGA DDS after immersion in Hank’s solution for 28 days (**a**) and cumulative Mg^2+^ content of Mg-based MAO-MT-PLGA DDS in leaching solution for 144 h (**b**). The red lines in (**b**) shows a sudden increase in Mg^2+^ release.

**Figure 4 pharmaceutics-14-01481-f004:**
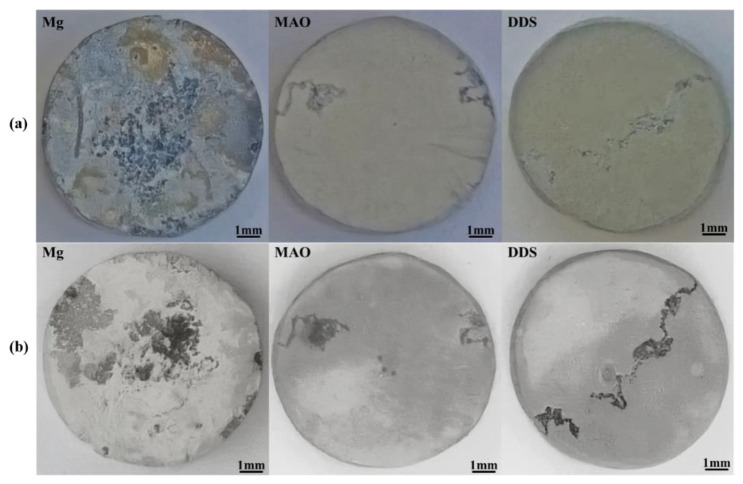
The surface macro morphologies of Mg alloy, Mg-based MAO alloy, and Mg-based MAO-MT-PLGA DDS before pickling (**a**) and after pickling (**b**) with the picric acid solution after immersion in Hank’s solution for 28 days.

**Figure 5 pharmaceutics-14-01481-f005:**
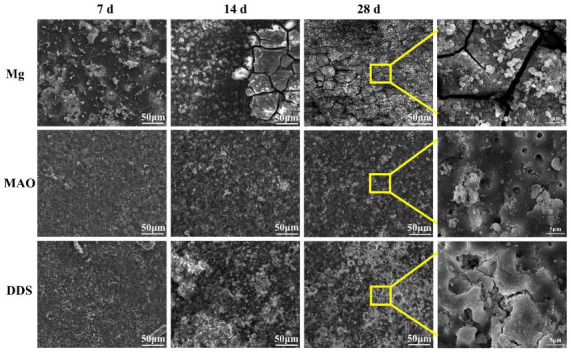
The surface micromorphology of Mg alloy, Mg-based MAO alloy, and Mg-based MAO-MT-PLGA DDS after immersion in Hank’s solution for 7 days, 14 days, and 28 days. The yellow squares shows enlargement of the marked area.

**Figure 6 pharmaceutics-14-01481-f006:**
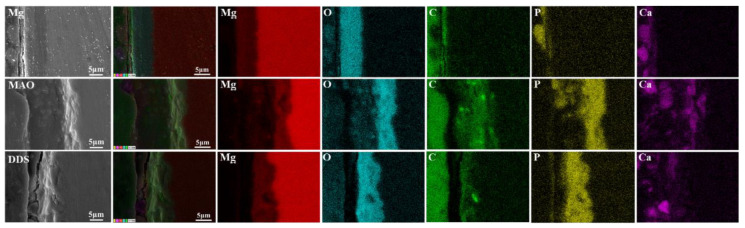
The cross-section images of Mg alloy, Mg-based MAO alloy, and Mg-based MAO-MT-PLGA DDS after immersion in Hank’s solution for 28 days.

**Figure 7 pharmaceutics-14-01481-f007:**
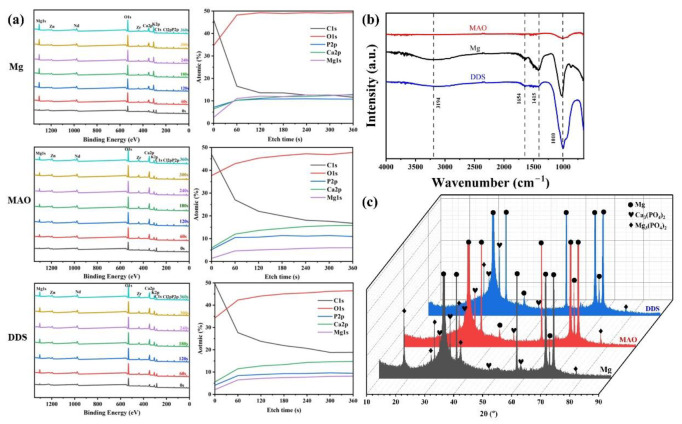
XPS patterns (**a**), FT-IR (**b**) patterns, and XRD patterns (**c**) of Mg alloy, Mg-based MAO alloy, and Mg-based MAO-MT-PLGA DDS after immersion in Hank’s solution for 28 days.

**Figure 8 pharmaceutics-14-01481-f008:**
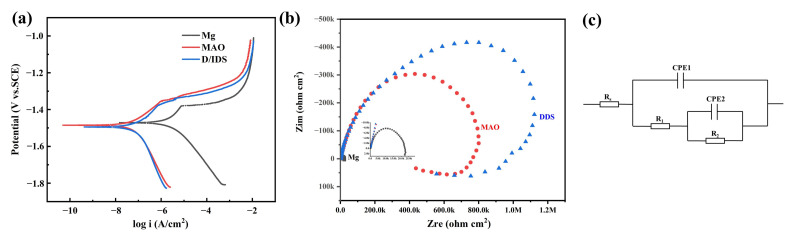
Potentiodynamic polarization curves (**a**), Nyquist plots (**b**), and impedance models (**c**) of Mg alloy, Mg-based MAO alloy, and Mg-based MAO-MT-PLGA DDS.

**Figure 9 pharmaceutics-14-01481-f009:**
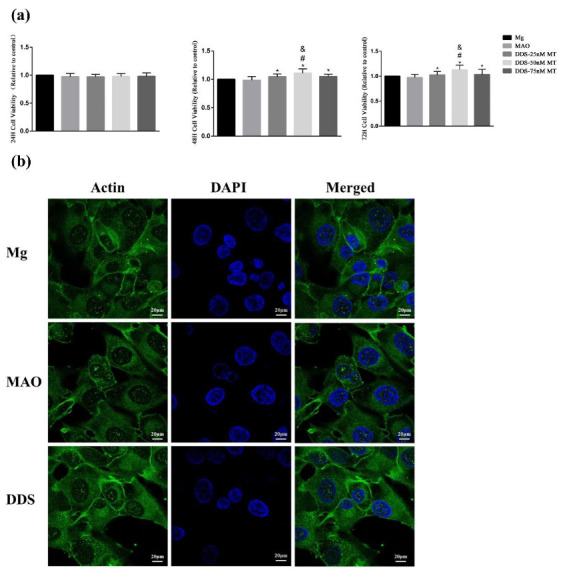
The cell viability of Mg alloy, Mg-based-MAO alloy, and Mg-based MAO-MT-PLGA DDS with three concentration gradients after co-culturing with the leaching solution for 24, 48, and 72 h. (**a**) and the cell adhesion and morphology (**b**) after co-culturing with the leaching solution for 24 h. “*” represents the DDS-50nM MT group compared with the control groups, “&” represents the DDS-50nM MT group compared with the DDS-75nM MT group, and “#” represents the DDS-50nM MT group compared with the DDS-25nM MT group.

**Figure 10 pharmaceutics-14-01481-f010:**
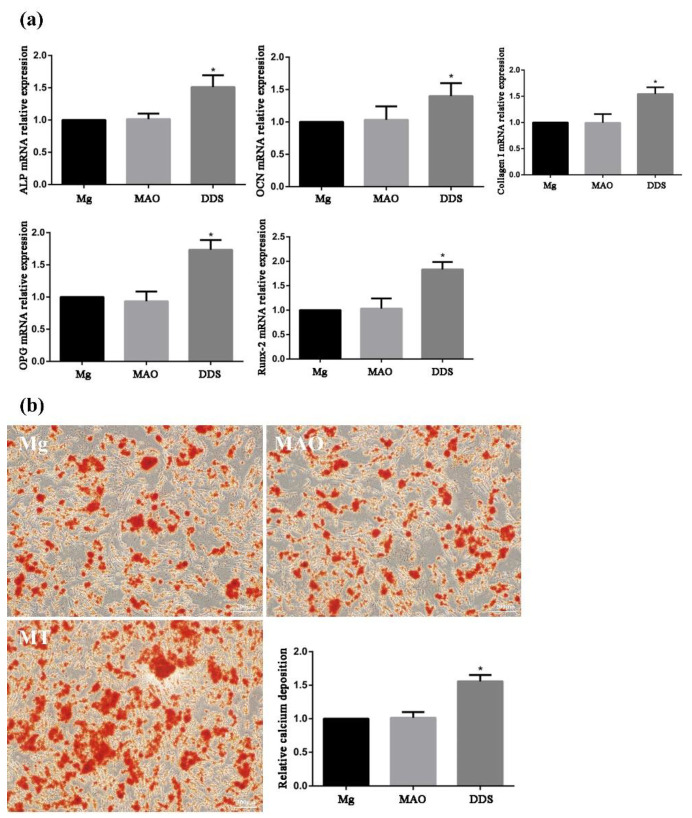
The mRNA relative expression of ALP, collagen I, OCN, OPG, and Runx-2 (**a**) by PCR tests and the relative calcium deposition by alizarin red staining (**b**) of Mg alloy, Mg-based-MAO alloy, and Mg-based MAO-MT-PLGA DDS. “*” represents the DDS group compared with the control groups.

**Figure 11 pharmaceutics-14-01481-f011:**
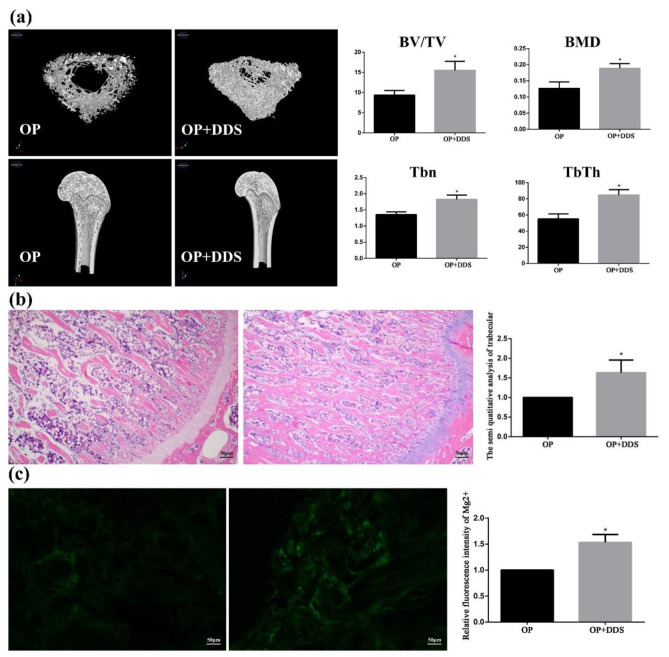
The 3D images of proximal tibia reconstructed by the Micro-CT (**a**), HE staining results of Trabeculae (**b**), and Magnesium content in bone tissue (**c**) of the treatment group rats and osteoporosis group rats. “*” represents the treatment group compared with the OP group.

**Figure 12 pharmaceutics-14-01481-f012:**
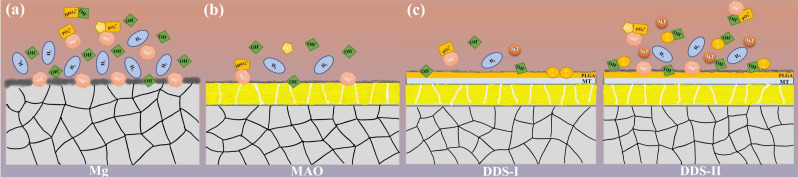
The degradation process and the release of Mg^2+^ and MT of Mg alloy (**a**), Mg-based-MAO alloy (**b**), and Mg-based MAO-MT-PLGA DDS (**c**).

**Table 1 pharmaceutics-14-01481-t001:** Primer sequences used in real-time PCR experiments.

Gene	Prime Sequence 5′–3′
ALP	F: AACATCAGGGACATTGACGTG
	R: GTATCTCGGTTTGAAGCTCTTCC
Collagen I	F: AGAGCTTCGGCAGCAGGA
	R: CTTATAGCAGTTCTGCCTGC
OCN	F: CACTCCTCGCCCTATTGGC
	R: CCCTCCTGCTTGGACACAAAG
OPG	F: GCGCTCGTGTTTCTGGACA
	R: AGTATAGACACTCGTCACTGGTG
Runx-2	F: CCTTCCAGACCAGCAGCAG
	R: TCCGTCAGCGTCAACACCA
β-Actin	F: GACAGGATGCAGAAGGAGATTACT
	R: TGATCCACATCTGCTGGAAGGT

**Table 2 pharmaceutics-14-01481-t002:** The results of Tafel fitting of potentiodynamic polarization curves.

Samples	Ecorr (V vs. SCE)	Icorr (nA/cm^2^)	CR (mm/Year)
Mg	−1.438 ± 0.036	1504.333 ± 504.155	8.317 ± 4.484
MAO	−1.482 ± 0.021	50.527 ± 11.323	0.289 ± 0.066
DDS	−1.521 ± 0.028	62.983 ± 4.406	0.283 ± 0.019

**Table 3 pharmaceutics-14-01481-t003:** Values of equivalent elements in impedance models of different samples.

Samples	R_s_ (Ω cm^2^)	R_1_ (Ω cm^2^)	R_2_ (Ω cm^2^)
Mg	30.92	546.3	21,714
MAO	69.32	878,860	31.09
DDS	89.7	828,510	500,060

## Data Availability

All data available are reported in the article.

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
