# Peer review of "A Degradable and Osteogenic Mg-Based MAO-MT-PLGA Drug/Ion Delivery System for Treating an Osteoporotic Fracture"

_pharmaceutics, 2022, doi:10.3390/pharmaceutics14071481_

Round 1

Reviewer 1 Report

The study title is focused on the degradable and osteogenic Mg-based MAO-MT-PLGA Drug/Ion Delivery System for treating the osteoporotic fracture. This is a very interesting topic to search possibilities to solve problems or improve fracture healing and management. There are important improvements that should be implemented before study can be reviewed.

  1. Please ask language reader/scientist to cure text. Please use official scientific nomenclature (example is line 328 “podynamic” or please read the first sentence from the abstract. Reader can be confused). This are only examples. Please cure the whole text.
  2. Please use advanced translator. There are grammar and language errors.
  3. Please try to use real and correct pictures connected with other data. Example is data from figure 11. Are there any fractures (line 397) visible in figure 11?
  4. Please add all limitations of the study.
  5.  Please rewrite references according to MDPI rules. References are not linked with the correct numbers, names are lost and additional text is entered.

Study plan is optimistic but data presented in not acceptable as the scientific work. Please cure text before review.

Reviewer 2 Report

"The manuscript assessed “Degradable and osteogenic Mg-based MAO-MT-PLGA Drug/Ion Delivery System for treating the osteoporotic fracture”. This research is under the scope of this Journal.   However, there are some concerns about the present manuscript:  
  • Correct typos in all manuscripts.
  (Abstract)  
  • Identified the aim of the study in the abstract. 
  • In the results, is important to show more information. Please add some of the p-values. 
  (Keywords)  
  • Please order the keywords / Mesh Terms alphabetically for a standardized presentation of the keywords.
  (Introduction)
  • What is the importance of this review study? Which results are comparable with other studies? What has this study been new? 
  • (Statement of Clinical Relevance)  
- Need to be reformulated, What is the importance of this study for the clinical?   (M&M)
  • How was the sample calculated? Did the authors perform a power analysis to evaluate if this sample size was appropriate? 
  • When mentioning materials or devices: for some of them, you don't mention the manufacturer at all, for some you mention only the manufacturer, for some the manufacturer and city, for some you mention the manufacturer and city/ country.  
  • After descrived the Statistical analysis used, Please Descrived the P-Value and Statistical Significance of this study.
  • How many operators performed the experiments? 
  • The word “ml” is correct to “mL”.
  (Results) - Improve the resolution quality of all figures and graphs (and a presentation). The font/language in the figure/caption is different from the text. Please, standardize the size and the font in the figures and charts with the font of the manuscript.    (Discussion)
  •  This material does not have porous? Can you discuss the importance of the porous or the space provision versus stabilization of the bone, please read this article, Palma et al. (2010, New formulations for space provision and bone regeneration. Biodental Eng. I, 71-76) reported the influence of different formulations of bone grafts in providing an adequate scaffold, thus emphasizing the importance of the type of carrier in the three-dimensional distribution of particles and also space provision in new bone formation. And 10.1002/jbm.b.34971.  
  • Please, identified more what was the strength(s) and limitations of this study? And also, implications for future perspectives in the bigger animal models, https://doi.org/10.1080/08941939.2016.1241840
(References)
  • Check the reference MDPI format in the manuscript and the references. The references have a different format one the manuscript presentation.”

Reviewer 3 Report

Degradable and osteogenic Mg-based MAO-MT-PLGA 2
Drug/Ion Delivery System for treating the osteoporotic fracture is significant scientifically with a need to revision

1-                  Some english errors must be revised like Podynamic polarizatiotention ? correct this word as potentiodynamic…..

2-                  You must add that as pH increase the passivation increases with time of immersion in the discussion.

3-                  Also, add the equations for the reaction of magnesium in neutral and basic medium.

4-                  Expect the mechanism of reaction with time.

5-                  The impedance model and data are not included in the article and should be added like in these references which are missed in the literature

1-  Suitability of Magnesium and Titanium alloys as implant materials, Materials Today: Proceedings18 March 2022.

2. (2019) Electrochemical, biodegradation and cytotoxicity of graphene oxide nanoparticles/polythreonine as a novel nano-coating on AZ91E Mg alloy staple in gastrectomy surgery, Materials Science and Engineering C, 103, 109780.

3. Micro Raman and XPS surface analysis to understand the electrochemical behaviour of AZ31 and AZ91 magnesium alloys as temporary implant materials. Materials Today Communications19 April 2022

4. (2014) Electrochemical impedance spectroscopy of chitosan coated magnesium alloys in a synthetic sweat medium, Surface and Coatings Technology, 283, 126 – 132.

     It is really important and interesting work.

Round 2

Reviewer 1 Report

Changes are accepted by reviewer.

Reviewer 2 Report

The authors improved the manuscript after the review.